
# 1 Earthquake-induced landslides monitoring and survey by means of
# 2 InSAR

Tayeb SMAIL[1], Mohamed ABED[1], Ahmed MEBARKI[2,3], Milan Lazecky[4,5]
[1]Department of Civil Engineering, Saad Dahlab University, Blida City, Algeria
[2]University Gustave Eiffel, Laboratory Multi Scale Modeling and Simulation (UMR 8208 CNRS/UPEC/U.Eiffel), 5 Bd
Descartes, 77454, Marne-La-Vallee, France
[3]Nanjing Tech University, 5 New Mofan Rd, Gulou, Nanjing, Jiangsu, Chine - Permanent Guest Professor within "High-
Level Foreign Talents Programme" grant
[4]IT4Innovations, VSB-TU Ostrava, 17, Listopadu 15, 70833 Ostrava-Poruba, Czech Republic
[5]School of Earth and Environment, University of Leeds, Leeds LS2 9JT, UK
*Correspondence to*: SMAIL Tayeb (st_gc@hotmail.fr)
**Abstract.** This study uses interferometric SAR techniques to identify landslides and lands prone to landslides, detect fringes
and changes in areas struck by earthquakes. The pilot study investigates the Mila region (Algeria) which suffered significant
landslides and structural damages (earthquake: Mw 5, 2020-08-07): the study checks ground deformations and tracks
earthquake-induced landslides. DInSAR analysis shows normal interferograms, with atmospheric contribution, and slight
fringes. However, it identifies many landslides, the most important (2.5 m displacement) being located in Kherba
neighborhood, causing severe damages to dwellings. In addition, SAR images and optical images (Sentinel-2) confirm site
investigations. Although in Grarem City, optical images could not detect any disorder, the DInSAR analysis detected some
coherence decays and small fringes (3.94 $Km^2$ area). These unnoticed ground disorders were confirmed during fields
inspection. Such results have key importance since they can serve as an alert to monitor the zone at the proper time.
Furthermore, Displacement time series analysis of many interferograms (April 2015 to September 2020) using LiCSBAS were
performed to investigate the pre-event conditions and precursors of the slopes instabilities., LiCSBAS detects a line-of-sight
subsidence velocity of -110 mm/y in the back hillside of Kherba, and high displacement velocity at specific points in Grarem
region.

## 25 1 Introduction

Although it is still challenging to predict exactly where and when natural hazards (earthquakes, landslides, floods, etc.) might
occur, the capacity to monitor and survey the zones prone to important landslides as well as the capacity to identify and locate
those impacted by earthquakes are key issues in risks mitigation, reduction, preparedness and adaptation. Actually, since
earthquakes and landslides might occur in many places worldwide, they might cause a huge number of victims, important
socio-economic, assets damages and losses. Their impact can be significantly reduced thanks to satellite imaging which allows
prediction and early alerts of some landslide cases (Jacquemart and Tiampo, 2021).
It is then worth detecting or predicting critical ground changes at specific places, either after a geotechnical disorder occurs
due to landslides and earthquakes mainly, or before it is suddenly triggered (Bakon et al., 2014; Galve et al., 2015). Such
challenges can be tackled by regular image processing oriented landslides areas monitoring, in the aftermath of earthquakes,
by means of SAR interferometric methods and optical images, for instance. Actually, since InSAR (Interferometric Synthetic
Aperture Radar) is an active sensor system that uses microwave signals to collect data backscattered from the earth's surface,





the use of satellite imaging systems InSAR appears as a cost-effective way for measuring millimeter-level displacements of the earth surface (Herrera et al., 2009), at a regional scale and can be used as an early warning system for the safety of structures and their surroundings (Galve et al., 2015; Roque et al., 2015).

The expected outcomes are based upon the processing of SAR data making use of Differential InSAR (DInSAR), Coherence Change Detection (CCD) and time series analysis of LiCSAR dataset using LiCSBAS software, to illustrate the advantages of high-resolution SAR sensing for the objective of tracking ground changes and landslides.

The performed SAR analyses can reveal some ground changes detected through DInSAR and CCD maps, as it is shown the illustrative purposes for the 7th of August 2020 earthquake (Algeria, Mila). These displacements, in the northeastern part at the Grarem City (2 km from Mila downtown) and Kherba City, caused a loss of coherence in CCD and fringes in DInSAR maps. Their extend affects areas of around 3.94 km$^2$ for Grarem and 2.1 km$^2$ for the Kherba landslide area. Furthermore, the time-series analysis using LiCSBAS reveals a slow deformation signal in the left part of Kherba landslide area.

The formed fringes and coherence loss in Grarem case indicate the boundary of a potential land failure. The results can serve as early warning information provided by the InSAR monitoring system, since the area should be monitored to investigate the existent or probable growing disorder.

## 2 Land and ground movements monitoring and surveying in the aftermath of an earthquake

### 2.1 Satellite images and methods - Case study

The present research study is multifold. It aims to use InSAR image processing for various purposes, in the case of landslides and earthquakes:

- Use the InSAR in the aftermath of an earthquake in order to identify the geotechnical disorders, their extent and locations. The Differential radar interferometry and the Coherence Changes Detection are the most adapted methods for ground and soil surfaces changes detection (Jung and Yun, 2020; Meng et al., 2020; Pawluszek-Filipiak and Borkowski, 2020; Tampuu et al., 2020; Tzouvaras et al., 2020). A city, Mila, in Northern Algeria, is considered as the pilot study. It has been struck by an earthquake in August 2020. The geotechnical disorders (landslides and surface faults) have affected significantly, during the same earthquake events series, two distinct zones being distant by almost 15 km from each other.

- Use the time series analysis to study the mean displacement and mean velocity before and after the occurrence of the main shock. For the city of Mila, the time series is performed out for a period extending from April 2015 up to October 2020, i.e. a long period before the (*April 2015 up to March 2020, i.e. 5 entire years*) and a period *4 months* ahead of the main shock in order to avoid a disturbance or bias that might be related to seasonal effects such as rains and vegetation effects (Lazeckỳ et al., 2020a).

- Compare and correlate the InSAR images processing results with the satellite optical images observations.

### 2.2 Pilot zone, earthquakes and landslides - Observed disorders

The case study area lies in Mila Province which is located in the northeast part of Algeria (Mediterranean zone), near the Dam of Beni Haroun. The Mediterranean zone is seismically active because of the northward convergence (4-10 mm/yr) of the African plate relative to the Eurasian plate along a complex plate boundary (USGS). Throughout the last years, several landslide events have taken place in the wider region of Mila. The seismic activities and landslides pose a persistent threat for built-up areas and facilities, such as roadways, bridges and tunnels, which need continuous monitoring and survey.

After an earthquake (Mw 5, 2020-08-07, epicenter 36.550° N - 6.271° E, Depth=10 km, (USGS)struck this region, important landslides were mostly observed in Mila City and its surroundings, see Figs. 1 and 2. Although, the earthquake was moderate, Beni Haroun Dam and the two large bridges built on the RN 27 highway needs to be inspected and their possible displacements monitored.

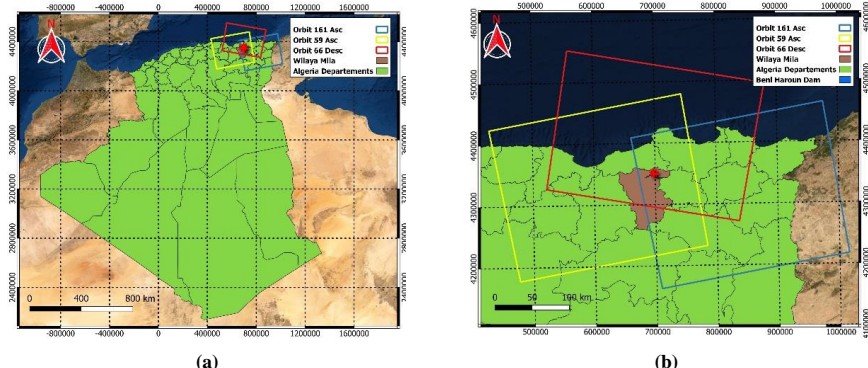

| (a) | (b) |

**Figure 1.** Mila location map (a), ascending and descending orbits footprints. Red stars indicate earthquake epicenter (QGIS, ESRI basemap).

In the present, two areas are studied, i.e. Kherba and Grarem Cities. The altitude at the top point 1 (Fig. 4.a) in Kherba hill is 654 m and 411 m a.s.l. in the upper point (2), with a horizontal length between 2.14 km and a slope of 11.34%. The maximum ground horizontal offset reached 2.5 m and the vertical deformations exceed 1.8 m (Fig. 2.b) at the top of Kherba hill (point A Fig. 4.a). The slope failure boundary of Kherba City is mapped as shown in Fig. 4.b. The Grarem area of interest (AoI) is located at east north of Mila in a hilly ground with an average slope reaching 12.5%, see Fig. 4.c.

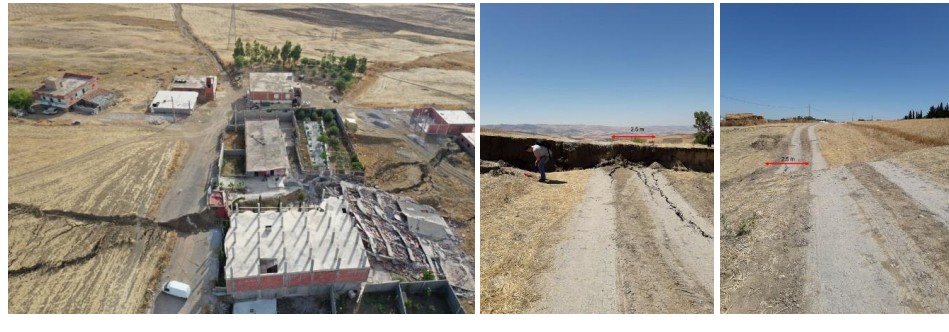

| (a) | (b) | (c) |

**Figure 2.** Ground cracks due to landslides in Kherba, Mila, ~2.5m offset towards the North, a. Drone aerial photo from (LNHC), b. & c.- Lateral displacements (*Photos: courtesy M. Yacoub A., University of Setif, Algeria*).

## 2.3 Pilot zone - Data and images collection

The dataset used for this study is collected from European Space Agency (ESA), via the Copernicus Open Access portal, and from the Alaska Satellite Facility (ASF DAAC). The C-band Sentinel-1 A and B, launched in 2014 and 2016 respectively, provide regular data sets (Morishita, 2021). For the InSAR use, the Interferometric Wide (IW) swath Single Look Complex (SLC) data is selected and processed with the open-source software SNAP (Sentinel Applications Platform). It is worth using data from many orbits to monitor the AoIs due to different oriented directions, incidence angles of satellites, and the ground topography. The optical images of Sentinel-2 sensors are obtained from ESA, whereas downloading and processing data is done via QGIS, Semi-Automatic Classification Plugin (SCP).

For Mila region, the AoI is covered by 3 orbits (2 ascending and 1 descending 66, 59, and 161) (Fig. 1). Since the present study intends to detect the areas influenced by landslides, many pre-event and post-event data are used in order to get an accurate evaluation of the event. Eighteen Sentinel-1 A and 17 Sentinel-1 B (total of 35) images downloaded for the period of 1 July 2020 to 26 October 2020, to monitor Mila's area, are used to perform out a detailed study on the land deformation and the dynamics of the landslide. Table 1 summarizes the appropriate interferograms with a small perpendicular baseline and short temporal baseline. Tables 1-2-2bis summarize the whole data collected for the case study.

**Table 1.** Characteristics of Sentinel-1 InSAR pairs for Mila case.



| IFG-ID | Track | M Date | S Date | $Bp$ [m] | $Bt$ [days] |
|---|---|---|---|---|---|
| IFG-1 | | 2020-07-22 | 2020-07-28 | -9.99 | 6 |
| IFG-2 | | 2020-07-28 | 2020-08-03 | 40.62 | 6 |
| IFG-3 | | 2020-08-03 | 2020-08-09 | -51.47 | 6 |
| IFG-4 | | 2020-08-03 | 2020-08-09 | -50.76 | 6 |
| IFG-5 | | 2020-08-09 | 2020-08-15 | -27.57 | 6 |
| IFG-6 | | 2020-08-09 | 2020-08-15 | 27.62 | 6 |
| IFG-7 | | 2020-08-15 | 2020-08-21 | -16.19 | 6 |
| IFG-8 | **66 ASCENDING** | 2020-08-21 | 2020-08-27 | 42.43 | 6 |
| IFG-9 | | 2020-08-27 | 2020-09-02 | -28.59 | 6 |
| IFG-10 | | 2020-09-02 | 2020-09-08 | 29.26 | 6 |
| IFG-11 | | 2020-09-08 | 2020-09-14 | 17.95 | 6 |
| IFG-12 | | 2020-09-14 | 2020-09-20 | -6.05 | 6 |
| IFG-13 | | 2020-09-20 | 2020-10-02 | -4.64 | 12 |
| IFG-14 | | 2020-10-02 | 2020-10-14 | 18.13 | 12 |
| IFG-15 | | 2020-10-14 | 2020-10-26 | -49.36 | 12 |
| IFG-16 | | 2020-07-27 | 2020-08-02 | 69.64 | 6 |
| IFG-17 | | 2020-08-02 | 2020-08-08 | -75.10 | 6 |
| IFG-18 | **59 ASCENDING** | 2020-08-08 | 2020-08-14 | -8.86 | 6 |
| IFG-19 | | 2020-08-14 | 2020-08-20 | 175.97 | 6 |
| IFG-20 | | 2020-08-20 | 2020-08-26 | -226.75 | 6 |
| IFG-21 | | 2020-07-22 | 2020-07-28 | -169.19 | 6 |
| IFG-22 | | 2020-07-28 | 2020-08-09 | 30.39 | 12 |
| IFG-23 | | 2020-07-28 | 2020-08-03 | 99.88 | 6 |
| IFG-24 | **161 DESCENDING** | 2020-08-03 | 2020-08-09 | -70.12 | 6 |
| IFG-25 | | 2020-08-09 | 2020-08-15 | 2.14 | 6 |
| IFG-26 | | 2020-08-15 | 2020-08-21 | 121.22 | 6 |
| IFG-27 | | 2020-08-21 | 2020-08-27 | -196.82 | 6 |

**Bt**: temporal baseline; **Bp**: perpendicular baseline.
All-time interval for InSAR pairs processing is 6 days, except the last three pairs of the 66 ascending pass that have 12 days.
Furthermore, since a bad coherence map of the IFG-24, may lead to misinterpretation of results, early images before the 3[rd] of
August with a time interval of 12 days are selected for the co-event interferogram for the 161 descending pass (IFG-22). The
gray cells in Table 1 represent the co-event interferograms. The perpendicular baselines guarantees also a good quality of
InSAR studies (Braun, 2019). As LiCSBAS time series analysis aims to investigate long period displacements and velocities
over a large area, 34 IFGs from the orbit 66 and 190 IFGs for the 161 ascending track (Table 2) are selected for the present
study.
**Table 2.** LiCSAR frames, analyzing periods and the total number of IFGs used in this study.

| Frame ID | Date | | Period | IFGs |
|---|---|---|---|---|
| | Start | End | | |
| 161A_05343_090806 | 2015-4-26 | 2020-9-26 | 66 month | 190 |
| 066D_05394_131311 | 2020-4-5 | 2020-9-26 | 6 months | 34 |


**Table 2bis.** Sentinel-2 optical images collected for the study case.

| Frame ID | Date | Duration days, to the main shock |
|----------|------|----------------------------------|
| Image 1 | 2020-07-30 | -7 days |
| Image 2 | 2020-08-09 | + 2 days |

**3 Methodology description and results**
Four aspects are investigated and compared in the present case study:

- The SAR Interferometric (InSAR) methodology, which is subdivided into three sub-groups:

- DInSAR for the phase changes (fringes),

- CDD for the coherence change detection,

- Time series analysis and LiCSAR data.

- The optical image processing.

**3.1 SAR Interferometric methodology**
The Interferometric Synthetic Aperture Radar (InSAR) is an active microwave imaging system. It is independent of sunlight
and can penetrate clouds, unlike optical imaging systems which are passive. InSAR uses the phase components of co-registered
SAR images of the same pixel to estimate the topography and to measure the surface change in the target area (Kim, 2013).
At least two constellation images are needed to generate an interferogram, which contains topographic, atmospheric effect,
baseline error, and noise components (Goudarzi, 2010; Kim, 2013; Netzband et al., 2007):
$\phi = \phi_{disp} + \phi_{flat} + \phi_{topo} + \phi_{atm} + \phi_{orbit} + \phi_{noise}$ (1)
Where $\phi_{disp}$ is the LOS displacement, $\phi_{flat}$ the flat earth phase, $\phi_{topo}$ the topographic phase, $\phi_{atm}$ is an atmospheric phase,
$\phi_{orbit}$ , the baseline phase and $\phi_{noise}$ is noise phase contribution (Kim, 2013).
The main steps for the study, by using SNAP software (DInSAR and CCD), are depicted in Fig. 3. It's worth notice that
for CCD processing, it is not necessary to follow the whole (DInSAR, Phase Unwrapping, and Phase to displacement).

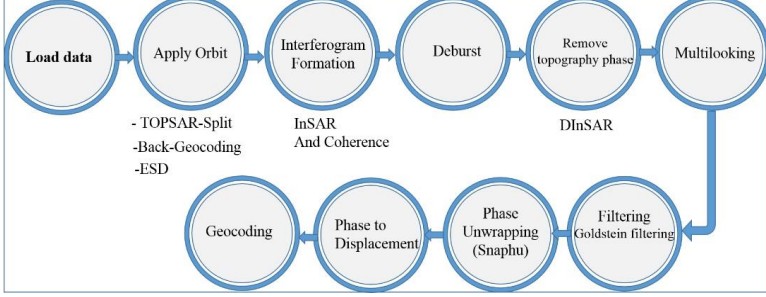


**Figure 3.** Workflow chart for DInSAR processing using (SNAP) software.
**3.1.1 Differential radar interferometry (DInSAR)**
Differential radar interferometry (DInSAR) exploits the phase difference to measure coherent changes or deformation between
two image acquisitions. It is often used for ground subsidence measurement(Canaslan Çomut et al., 2020; Galve et al., 2015).
One of DInSAR's limitations is that the changes are not measurable in case of non-coherent events (e.g., rapid landslide)
(Braun, 2019).
**3.1.2 CCD Times series analyses**

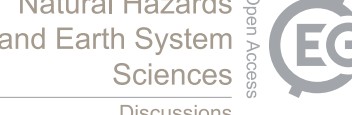
The estimated coherence is considered as a quality indicator of an interferogram (Jacquemart and Tiampo, 2021). Actually, it
indicates that the phase and amplitude of the received signal express the degree of similarity between the images pair. The
pixel coherence **γ** of two SAR images is estimated on the basis of N neighboring pixels (Jia et al., 2019; Wang et al., 2018).
$$\gamma = \frac{\sum_{i=1}^{N} S_{1i} S_{2i}^{*}}{\sqrt{\sum_{i=1}^{N} |S_{1i}|^2 \sum_{i=1}^{N} |S_{2i}|^2}}$$
(2)

Where: $S_{1i}, S_{2i}$, are the complex signal values of the SAR image pair, N is window of neighboring pixels, $*$ is the complex
conjugate.
The coherence values range between 0 and 1 so that the map is represented as a gray color which 0 is white and 1 is black.
**3.1.3 Time series analysis and LiCSAR data**
The LiCSAR system processes Sentinel-1 InSAR datasets, and generates wrapped, unwrapped interferograms and coherence
maps (Lazeckỳ et al., 2020b), with a final product resolution of ~26.5 m (Lazeckỳ et al., 2020a). For such purposes, the open-
source LiCSBAS software, adopted in the present study, is used for InSAR time series analysis based on LiCSAR data. It is
able to generate maps of mean velocity displacement and deformation time series for all processed frames. Furthermore, it is
easy to implement and does not request high-performance computing facilities (Morishita, 2021).
In addition, the mechanism of landslides can be thoroughly studied through LiCSBAS analyses. They rely on the InSAR
time-series analysis package integrated with Looking into Continents from Space with Synthetic Aperture Radar (LiCSAR),
(Lazeckỳ et al., 2020b). Such time series analyses are very helpful in identifying, for a given landslide or geological disorder,
the prior patterns of ground movements versus the time as well as foreseeing a potential disorder.
**3.2 Optical image processing**
The optical methods are a passive detection way that needs sunlight and clear weather conditions in order to exploit the data.
The optical data collected from the ESA platform (Sentinel-2) is treated and plotted using QGIS, in the present study.
**4 Application to the case study and results**
The case studies are located in two different sites and both areas of interest are located in Algeria. They have a hilly relief: the
first one is located northeast of Mila City (Grarem) and the second is at the west part of Mila City (Kherba). To monitor the
AoIs, several available images are processed and used with different orbits directions (Ascending and Descending, see Fig. 1),
in order to catch deformation from different angles along the sensor's LOS. The InSAR technique is used in both areas, in
order to detect land deformation and landslides caused by the earthquake.
The present study uses Sentinel-1 A and B datasets: The Sentinel-1 sensors have a wavelength of 5.546 cm (ESA) and are
right side-looking with an incidence angle ranging approximately from 20° to 46° (ESA, 2012), which are suitable for change
detection and the monitoring of large areas. Furthermore, optical sensors data from Sentinel-2 are used in order to validate the
ground changes detected by InSAR.
The four adopted methods are applied for Mila case study in order to:
- detect and measure the co-event surface displacements and landslides, caused by the earthquake (CCD and DInSAR)
- monitor their dynamic evolution in the first weeks and months, at the post-event period (CCD and LiCSAR data)
- analyze their possible initiation ahead of the earthquake by months and years, at the pre-event period (Time-series
methods and LiCSAR data)
- corroborate the results by comparing several methods outputs, i.e. SAR (CCD, DInSAR, LiCSAR), aerial optical photo
(Sentinel-2), and field surveys.





The quality of the SAR image is consistent with the topography slopes and area roughness. Actually, the AoI has rough
topography, hills, and rivers. Selecting either ascending or descending passes, relying on which will avoid some limitation of
InSAR is an extremely essential action able to infer the deformation from various angles. Therefore, considering the regional
topography and geology of the AoI is necessary to process InSAR and results interpreting.

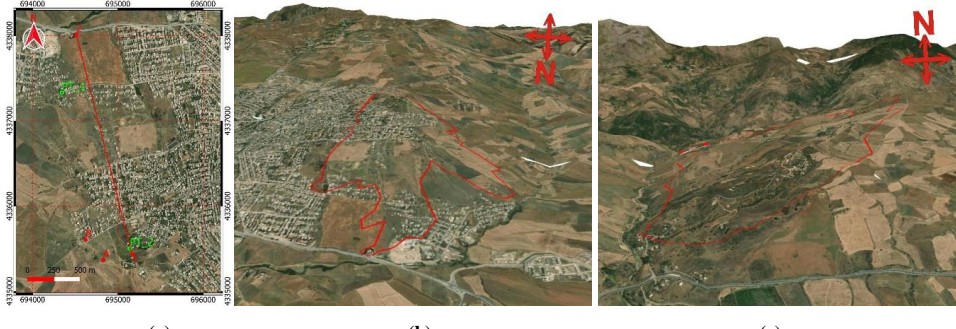


**(a)**        **(b)**        **(c)**

**Figure 4.** 3D view of AoIs, Kherba AoI and Grarem using QGIS with DEM SRTM 1sec and ESRI basemap, a & b are Kherba AoI, c is
the Grarem case area, the red polygon is the boundary of change detected by InSAR.
Differential InSAR (DInSAR) method is helpful to investigate co-seismic effects and detect changes in the ground. The
produced Interferograms and coherence images are projected to WGS84 reference, with a pixel size of 13.4 m. The unwrapped
interferograms present phase contribution of many noise resources (atmospheric), see Fig. 5. In general, strong earthquakes
cause large-scale fringes patterns around the epicenter which is not the case in the event under study (a moderate earthquake).
Processing DInSAR analysis may then lead to misinterpretation due to atmospheric contribution in differential phase
interferograms (Fig. 5). In the study case, no regional deformation due to the earthquake is observed and there is no need to
continue investigating the dam and the two bridges by simple DInSAR. However, this moderate earthquake has triggered small
deformation and landslides in Grarem, Kherba, and Azeba, see Figs. 7, 10, and 20.

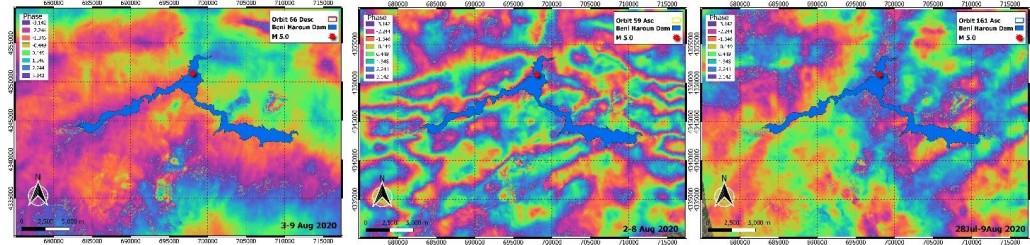


**Figure 5.** Wrapped Interferograms from Sentinel-1 for IFG-3+IFG-4, IFG-17 and IFG-22, The red star is the epicenter location USGS.

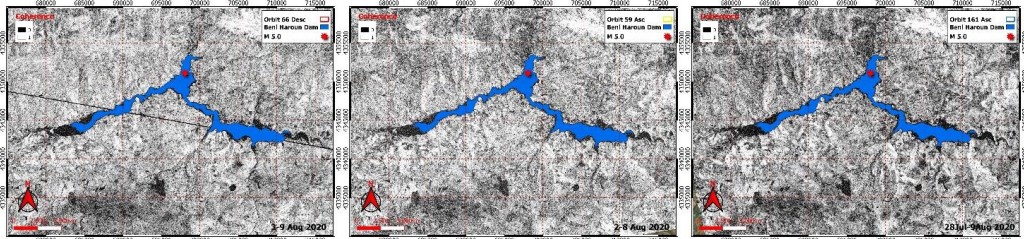


**Figure 6.** Mila Area, InSAR coherence maps for IFG-3+IFG-4, IFG-17 and IFG-22.
The IFG-3 and IFG-4 are merged in one image due to the AoIs, which are located between two different image acquisitions
in orbit number 66. In order to monitor the dam and bridges, it is highly recommended to use PS-InSAR for regional and local
ground deformation detection (Hooper et al., 2004; Rapant et al., 2020; Sanabria et al., 2014).
**4.1 Case of GRAREM**



The detection of deformation or changes between two InSAR images reveals a small change in the region of Grarem. This
change is detected as small fringes, each fringe corresponding to a displacement of a half-wavelength (λ=5.546 cm) in the LOS
direction (Fig. 7). Usually, coherent change does not appear in coherence images as dark region, but in the study case, the
outer borderline of the fringes region shows incoherence change which is clearly visible in coherence maps (Fig. 8).

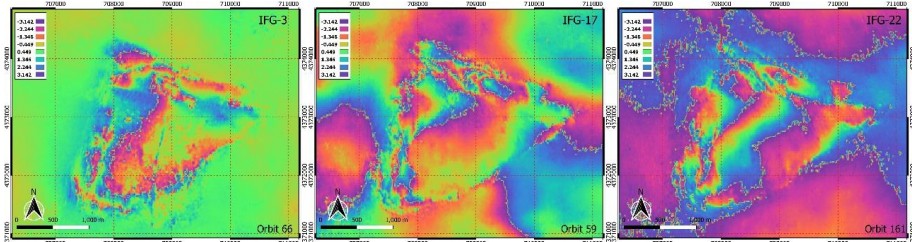


**Figure 7.** Detected fringes in Interferograms N° 3, 17 and 22, focus images on Grarem zone.
A time series analysis needs then to be performed out to prove whether this contour was formed at the event occurrence
date (August 7, 2020). The coherence maps present a dark polygon which is related to incoherent change or deformation. But
inside the AoI, the results show a coherent change which means that this area has deformed as a block up or down.

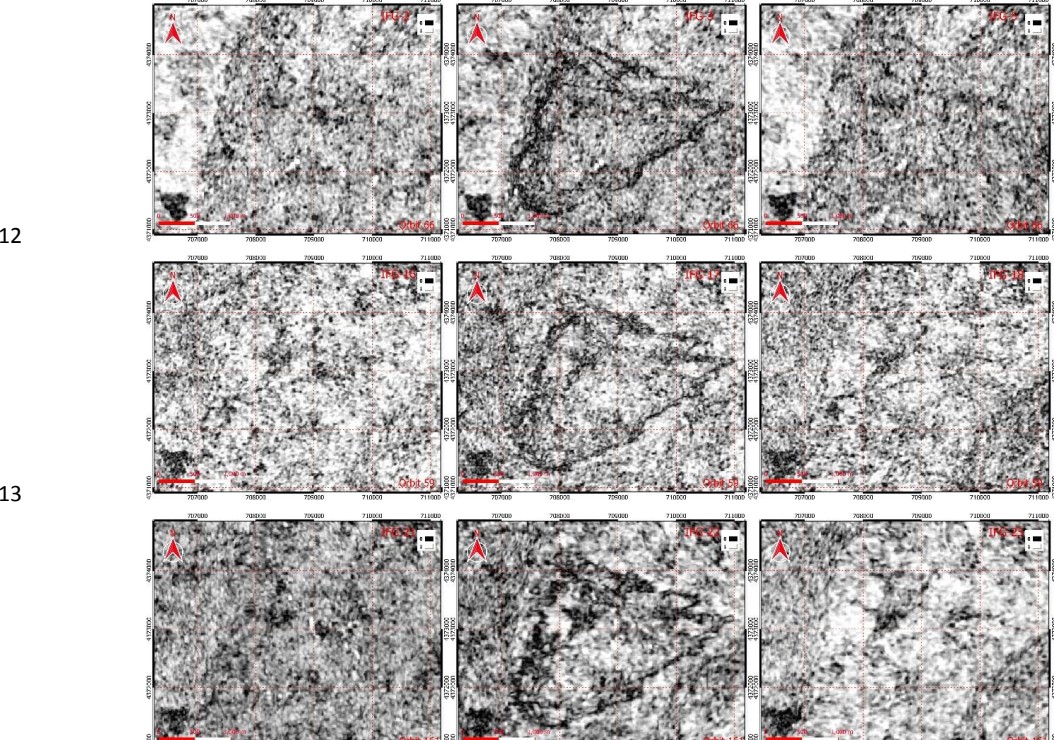




**Figure 8.** Coherence maps of Grarem AoI: the images represent pre-event, co-event and post-event for each orbit.
The affected area covers about 3.94 km² as estimated from phase and coherence maps, with an average runout distance from
the top to downhill of 2.6 km.

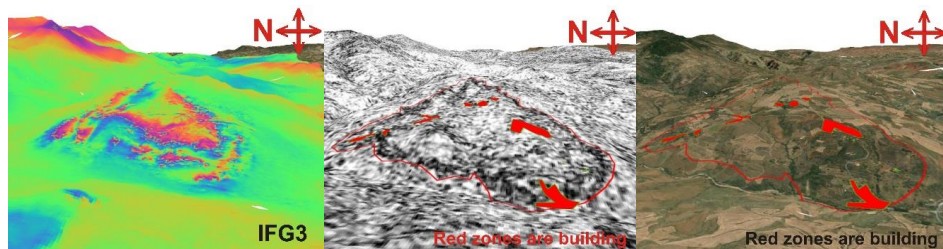

**Figure 9.** 3D view of Grarem Area, Images of IFG-3. Each fringe = wavelength/2 in LOS, and red zones represent existing building compounds (QGIS, ESRI basemap).

**4.2 Case of Kherba**

DInSAR is expected to be more suitable for slow and gradual movements (Cascini et al., 2013). In the present study, Kherba's landslides exceed the capabilities of DInSAR since this method cannot measure the changes due to incoherent change at the first event. Phase images of the Region of Interest (RoI) show a clear decorrelation and consequently, the phase information is no longer convenient for analysis.

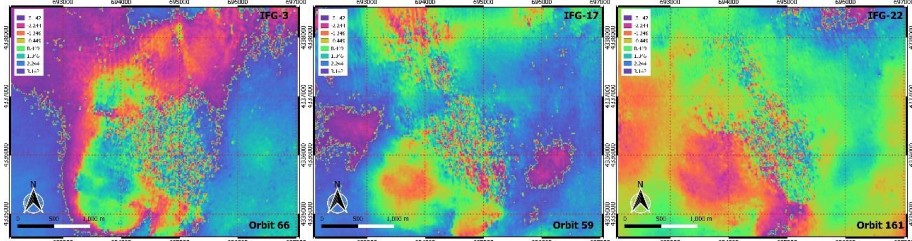

**Figure 10.** Kherba main event interferograms, biased pixels in the RoI correspond to changes.

In such cases of incoherent changes, we can serve to the coherence change detection (CCD) method which is suitable and able to monitor the event.

**4.2.1 CCD Times series analyses**

For the case study, the coherence maps (Figs. 11-13) show very low coherence in the area (RoI) and indicate that some changes have occurred. This may confirm whether the decrease of coherence values is due to the hazard or it is naturally low. The Coherence Change Detection (CCD) is useful when the change is incoherent in the scene, since CCD quantifies changes between two SAR images, and is represented as a decay of coherence values. To distinguish between natural low coherence (e.g., water, vegetation…) and induced surface changes, a second coherence map is needed in order to serve as a reference to be compared with pre-event images. It is preferable to mask out the rest of the non-changed area with a ratio of pre-event by co-event image and filter values that are equal of less than 1 (see Fig. 13).

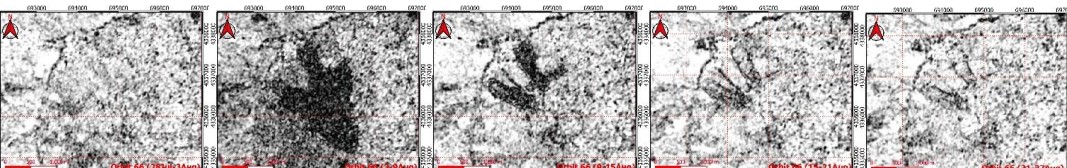



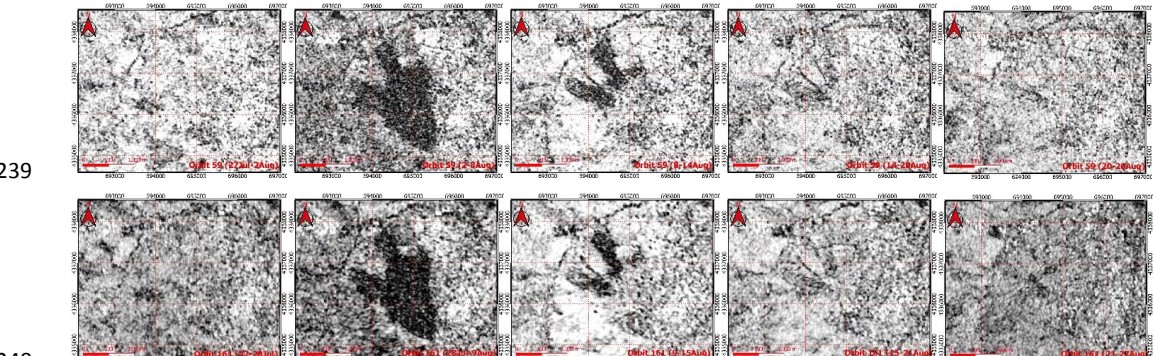



**Figure 11.** Coherence time series maps of Kherba, Sentinel-1.

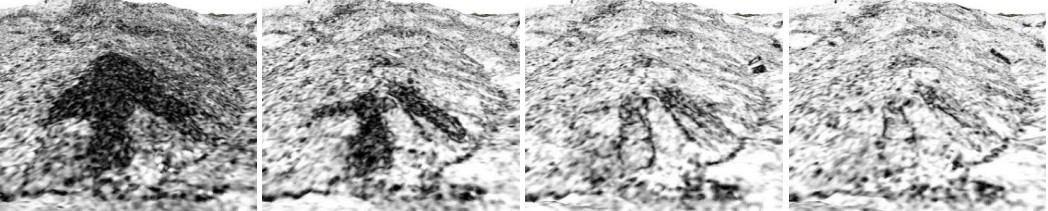


**Figure 12.** 3D view of Kherba City, Sentinel-1 Coherence images Orbit 66.
The CCD time-series display the changes in the AoI over time for Kherba. The dark region represents the main changes
that occurred during the co-event period. The landslide shape is divided into two toes at the lower side of the Hill, as shown in
Figs. 11 and 12.
During the first week following the earthquake, changes are detected in the lower side of the hill, and lasted until the late date
of August 2020 (IFG-8 orbits 66, IFG-27 orbit161 and IFG-20 for orbit 59). Afterwards, many other sources of noise were
present in the AoI, which makes this technique less efficient (weather, human activities). Most of the processed images are 6
days' intervals, except the orbit 161 in which the co-event image (IFG-24) was not enough good to compare with other pre-
post event images: it is then replaced by the IFG-22. Figs. 14 and 15 illustrate how the image selection may change the
interpretation of results.
To quantify the change, an RoI is represented in Fig. 13, and the plots in Figs. 14 and 15 shows also the calculated average
coherence values and the decreased percentage values inside the RoI. For the 66 orbit's pairs, the RoI average coherence starts
by 0.66 during the pre-event period (IFG-2) and decreases to 0.51 during the co-event period (-31%) (IFG-3). For orbit 59
pairs, it decreases by 27% with a mean value of 0.60 (IFG-17) after an initial mean value of 0.77 (IFG-16).

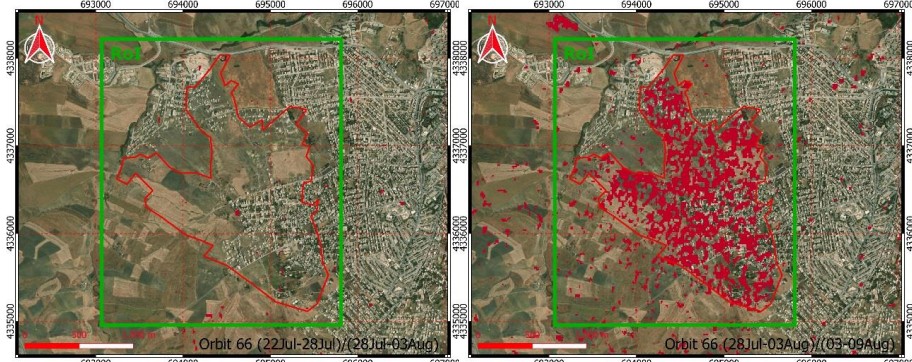


**Figure 13.** Pre-event coherences ratio (left), coherences ratio (right), Sentinel-1 Orbit 66. The green box indicates the scope of the RoI, red
spots represent significant changes of coherence in the landslide region (QGIS, ESRI World Imagery basemap).



The last orbit 161 pairs make an exception due to an initial bad coherence map (IFG-23 and 24), see Fig. 14.c. However, the
previous pair (IFG-21 and IFG-22) gave a value of 0.57 (IFG-21) which decreased to 0.52 (IFG-22), i.e. 11% of change.
**Table 3.** Mean coherence change values inside the ROI.

| Orbit | Pre-event coherence mean | Co-event coherence mean | Change |
|:---:|:---:|:---:|:---:|
| **66** | **28Jul_03Aug** | **03_09Aug** | -31% |
| | 0.66 | 0.51 | |
| **59** | **27Jul_02Aug** | **02_08Aug** | -27% |
| | 0.77 | 0.60 | |
| **161** | **22Jul_28Jul** | **28Jul_09Aug** | -11% |
| | 0.57 | 0.52 | |

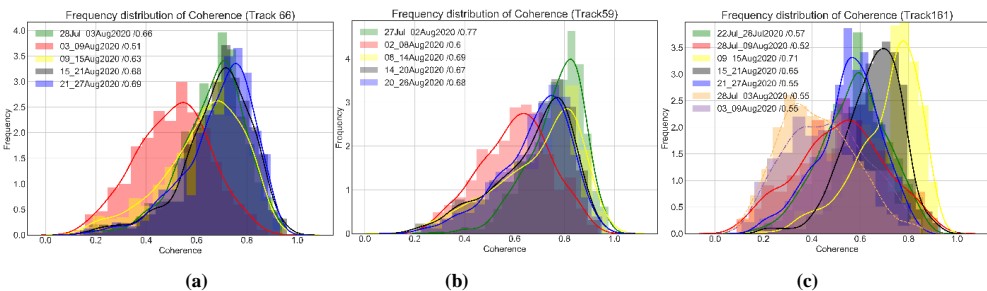


**(a)**                                    **(b)**                                    **(c)**

**Figure 14.** Frequency distributions of coherence values within RoI for coherence time series images.

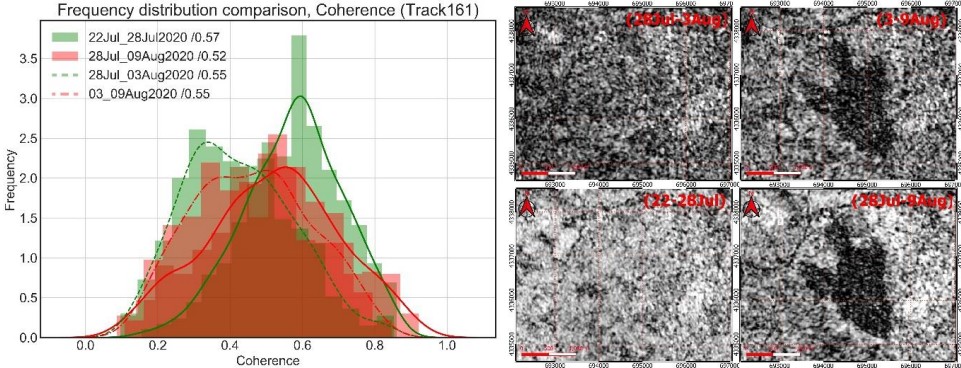


**Figure 15.** Effect of bad coherence shift: the cause is the acquisition of 3rd August start at (17:28:15, orbit 161) under bad weather
conditions in the acquisition time according to precipitation site (WWO) (rainfall in that daytime), compared to the acquisition of the same
day but not the same time (05:37:58 for orbit 66).

The surface area covers 2.1 km², derived from the coherence images, and the shape ends by two toes. The runout distance

is 2.4 km for the right toe and 2.15 km for the left one. The CCD method has the potential to differentiate between the areas
impacted by induced changes and those affected by other sources of noise. The ratio operation is useful in canceling out other
noise factors and improve the detection of changes in the region.
**4.2.2 Optical detection**
To validate the SAR methods results, two images from Sentinel-2 are downloaded and treated, the images being dated 2020-
07-30 (a week before the main shock) and 2020-08-09 (two days after the main shock), and the optical data is treated using
QGIS. The passive detection shows that an important ground surface displacement affected the Kherba neighborhood, over an
area of 1.32 km². The landslide shape of deformation has only one toe at the lower part of the hill compared to the CCD method
results.


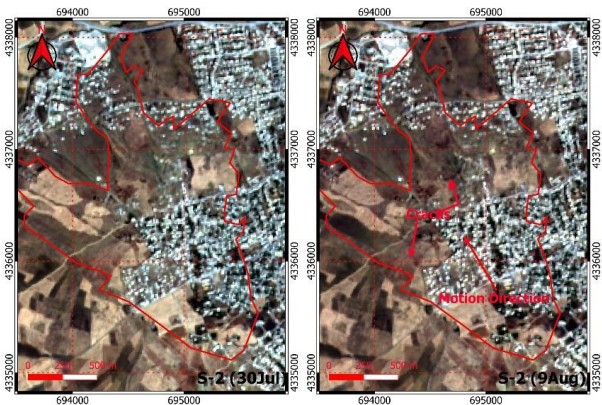



**Figure 16.** Sentinel-2 Optic Images: (a)- dated 30-Jul 2020, (b)- dated 09-Aug 2020.
**4.3 LiCSBAS analyses**
LiCSBAS is an open-source program for InSAR time series analysis based on LiCSAR data. The LiCSAR system
automatically processes Sentinel-1 InSAR datasets (Lazeckỳ et al., 2020b), to generate wrapped, unwrapped interferograms
and coherence maps (Lazeckỳ et al., 2020b), with a final product resolution of ~26.5 m (Lazeckỳ et al., 2020a). LiCSBAS
exploits the data of LiCSAR in order to generate maps of displacement mean velocity and deformation time series plots
(Morishita, 2021).
Displacement time series and velocities analysis of the region is performed out using LiCSBAS. It allows identifying
whether unstable conditions pre-existed or are still undergoing. The study started from the 5$^{th}$ of April to the 26$^{th}$ of September
2020, for the orbit 66 and from 26 April 2015 to 26 September 2020 for the orbit N° 161.

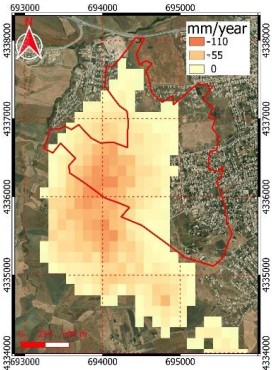


**Figure 17.** Line of sight (LOS) displacement velocity map, red line is landslide area (066D_05394_131311) (QGIS, ESRI basemap).
The time series detected subsidence at the west part of Kherba. This region is on the other hillside of Kherba Hill. A site
investigation did not find any drilled wells. One may assume that this subsidence is not due to any pumping of groundwater.
Therefore, another possible explanation is probably related to the mass movement of the main landslide hillside. The
displacement velocity in Kherba is about 110 mm/year, Fig. 17.
For the Grarem case, the velocity map looks stable between the same dates (April 5 to September 26, 2021). The change
occurred rapidly and is removed by the filters. For illustrative purposes, the displacement time series of some points are
illustrated in Figs. 18 and 19.

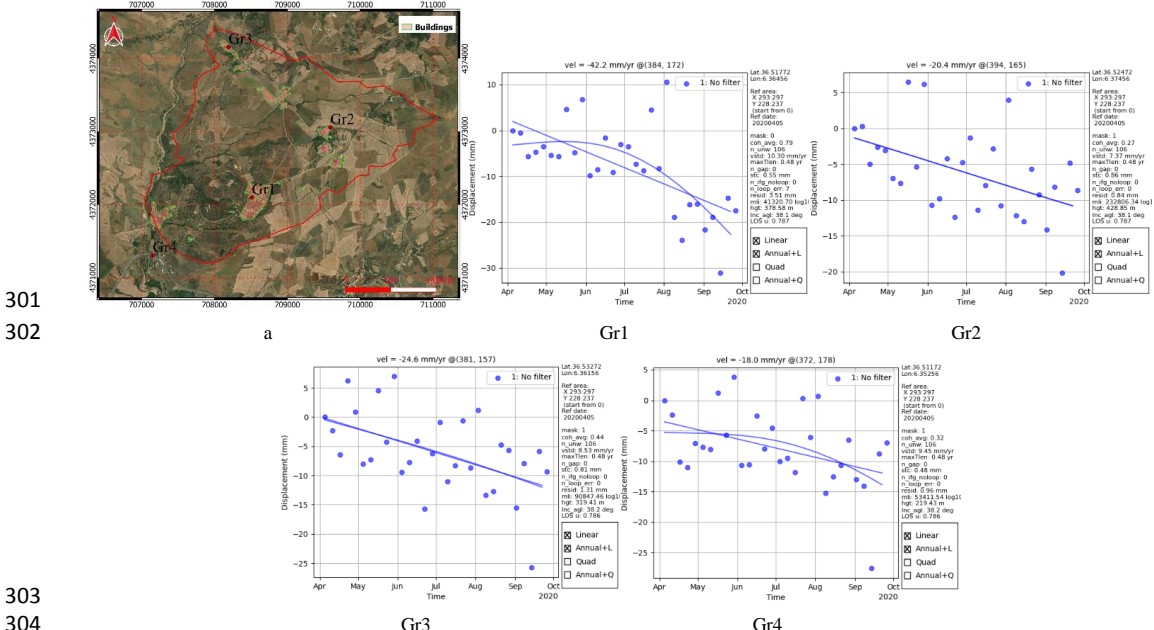


a                         Gr1                          Gr2

Gr3                          Gr4
**Figure 18.** Displacement time series (orbit 66), corresponding to the points in image (a) the displacement is relative to the reference point,
(a) Grarem region, (QGIS, ESRI basemap).

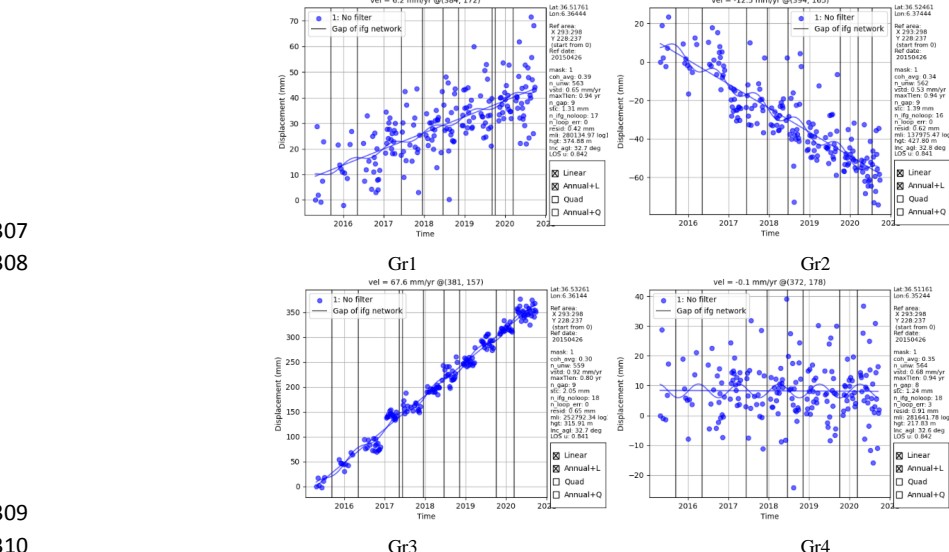

Gr1                                          Gr2

Gr3                                          Gr4
**Figure 19.** Displacement time series (orbit 161), corresponding to the points in Fig. 18.a, displacement is relative to the reference point.
During the LiCSBAS processing, a primary stable reference point is selected (36.455885° N, 6.276909° E). This method
proves to be efficient for large-scale deformation monitoring and slow coherent changes.
**4.4 Discussion**
InSAR monitoring proves its ability to detect land changes. First, landslides and land deformation can be detected remotely by
InSAR. Furthermore, optical images could detect only one case (Kherba). The theoretical results were validated by site visiting
and investigation, i.e.:



- Compared to results obtained from optical for the Kherba landslide, InSAR is more precise for detecting small
deformation (2 toes in CCD maps). Besides, the other techniques did not detect the full changing area in the region (only
one toe).

- Landslides of this magnitude exceed the capabilities of DInSAR, and their extreme loss of coherence. The interferograms
of co-events are strongly decorrelated. Therefore, the phase information is no longer usable and one cannot measure the
displacement of incoherent ground changes (Landslide).

- Land deformation in Grarem first detected by DInSAR, was confirmed by a site visit, during which small cracks were
visible on the ground (incoherent boundary region). Due to incoherent boundaries and because the displacement is
probably larger than what can be measured by one interferogram (depending on the wavelength, 5 cm for Sentinel-1), the
deformation measurements in this case are not reliable and accurate.

- Another landslide detected by InSAR in the Azeba region (6 km east of Mila) was visited too: the area covers 0.42 $km^2$
and the site investigation (Fig. 20) confirms the landslide.

- Analysis with LiCSBAS revealed new hillside deformation (subsidence) which is probably a consequence of the mass
that moved in the main landslide hillside. Displacements time series, in Grarem region at some point, show deformation
along LOS with velocities ranging from 6 mm/year to 67 mm/year. This method is preferable in the large-scale area and
large-period analysis.

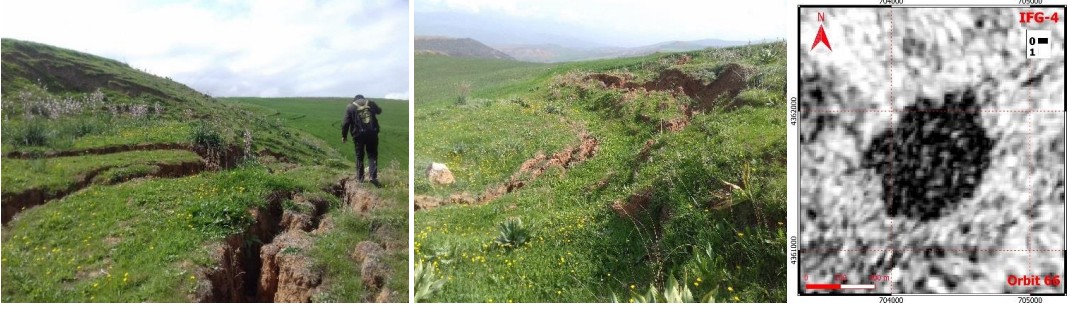

**(a)**                   **(b)**                **(c)**
**Figure 20.** Landslide occurred in Azeba region (2 km North) detected by InSAR: *(a) & (b)- visible ground cracks, (c)- coherence map of*
*Azeba zone delimited by the cracks.*
**5 Conclusions and Recommendations**
In this paper, active and passive space-based satellite data are used to monitor and study the impact of natural hazards
(earthquakes and landslides) on struck areas. The C-band Sentinel-1 SAR datasets (active sensing) and optical images of
Sentinel-2 data (10 m spatial resolution) were used in this study to investigate the area, the passive images were used to validate
the active results. For the InSAR processing, the use of DInSAR, CCD methods and the LiCSBAS tool has been able to
generate a detailed time series analysis of ground changes.
InSAR techniques can extract useful geodetic information, such as the ground movement and track surface deformation over
large areas with centimetric accuracy in coherent change cases. The present research study has demonstrated that the InSAR
processing is adapted to study earthquake and landslides zones. As a result, three primary land failures were detected over the
study area using InSAR.
DInSAR is poorly suited to track and detected landslides. It is represented as a pixel decorrelation in phase interferograms and
high decay in coherence values. CCD is further suitable to map earthquake-induced landslides that may remain undetected
using coherent methods (DInSAR). The estimation of their horizontal displacement is a challenge to be inferred.
The Grarem deformation looks as a landslide that has just been initiated, but might extend under an upcoming triggering event.
Actually, the failure plane rim is presented as a dark line in the coherence map or as the circumference of the fringe in phase
maps (estimated area 3.94 sq. km). This impending land failure needs therefore a thorough and real-time monitoring and



adequate geotechnical studies. PS-InSAR can serve as an efficient and low-cost monitoring method able to obtain millimeter-
level precision displacement measurements over selected points in the area (Jia et al., 2019).
It is worth to increase awareness of possible future geotechnical threats in a timely manner, through on-site monitoring using
GPS, crack meters, and by placing inclinometers in the Grarem area, in order to develop a model of the slope stability.
**Acknowledgments:** In this work, we used SNAP and QGIS to analyze and plot maps. ESA, Copernicus, and COMET for
providing Sentinel data. The authors are grateful to European Space Agency (ESA) for providing freely the data through
Copernicus Program.
**Conflicts of Interest:** The authors declare no conflict of interest.

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
