# Peer review of "Earthquake-induced landslides monitoring and survey by means of"

_Natural Hazards and Earth System Sciences, 2021_

## Author Response (AR1)

| Block | Referee's comments (1) | Authors' replies and comments |
|---|---|---|
| I | The authors show the application of the differential interferometric technique to evaluate the ground displacements induced by an earthquake. The images processed by the SNAP software application are SENTINEL-1. Finally, they validate their findings with SENTINEL-2 optical image analysis. | *General comments from the authors* ➔
The purpose of the paper is multifold i.e.:
- Identify the extend of landslides and geotechnical disorders caused by an earthquake. For this purpose, the methodologies used in the paper are: coherence change detection and phase changes (DInSAR), optical images (Sentinel-2), "*historic*" data processing (LiCSBAS).
- Investigate the accuracy and validity of such identification. The case study concerns regions and seasons during which there was little vegetation and rain. The main event that caused the landslides and geotechnical disorders is an earthquake Mw5.0 that struck the northeastern part of Algeria (Mila city, August 2020). |
| II | The work has several shortcomings in all its parts and is poorly written. All sections are in need of thorough revision. Some of them could be deleted as they have only two lines. | The authors will thoroughly revise the paper upon request from the Editors. Obviously, the revised paper will take into account the whole comments and remarks raised by the referees and readers. |
| III | Some software is introduced without explaining its usefulness (LiCSBAS, LiCSAR). | LiCSBAS and LICSAR are detailed in references. [ Lazeckỳ et al., 2020b and Morishita, 2021]. They have been used in the present paper in order to processed available images collected during two duration periods:
- Long period i.e. 5 years (2015-2020): in order to detect any previous gradual displacements or disorders in the region.
- Short period i.e. 4 months before the main shock and two months after the main shock: in order to analyze the velocity of the changes and disorders.
The analysis for the short period, i.e. the near-event period, was able to detect and catch the subsidence. |
| IV | Declaring to use SENTINEL-1 images, even reporting tables, the authors do not clarify which images they actually used. The whole iconographic part is illegible and unclear. | Table 1 presents all the images used in the study and they are labelled as IFG-ID, Orbit, and their dates. For the results section, every image contains the description of its source, by IFG-ID or by the image's dates. |
| V | The results are presented in a confusing manner. | The revised version, upon request of the referees and editors, will modify the structure of the text in order to present clearly the main results, i.e.:
- The Coherence Change Detection and Phase Change were able to detect the extent of the zone that suffered important landslides and geotechnical disorders during the main shock. Two important zones have been identified (Kherba and Grarem)
- The optical images were also able to identify the landslide extend and disorders in Kherba, in which the mean horizontal displacement reached 2.5m. These optical images processing were in accordance with the CCD analysis in terms of zones affected by the disorders and landslides.
- The optical images were unable to detect the disorders in Grarem in which there was no landslide although there were a lot of ruptures and cracks. However, a field inspection has confirmed the results of the CCD and DInSAR analysis in terms of pattern and limits of the zone affected by the disorders (surface rupture).
The analysis of InSAR images (using specific software, namely LiCSBAS) for the short period, i.e. the near-event period, was able to detect and catch the subsidence in the case of Kherba where the landslide was important. |
| VI | The citations used in the context of interferometry theory are inadequate as they do not take into account historical works (Hanssen, 2001, Franceschetti et a., 1992, Gabriel et al., 1989....). | Actually, InSAR is widely used, with related developments and works detailed in many articles and books, which we can cite in the bibliography, for the revised version, as suggested by the referee. |
| VII | Validation with optical data is practically absent. | The validation with optical data is commented in Block V (see Grarem and Kherba) in which the field inspection and optical processing of Sentinel-2 data (Figure 16 dated 30-Jul 2020 and 09-Aug 2020) illustrate the change that occurred in these zones. |
| VIII | Having said this, I believe that the work should be rejected. | The authors are respectful to the Editors and Referees' decisions and recommendations, as well as the readers' comments. Hopefully, Editors and the Referees will give a chance for a revised version before possible acceptance. |

| Block | Referee's comments (2) | Authors' replies and comments |
|---|---|---|
| I | The paper "Earthquake-induced landslides monitoring and survey by means of InSAR" presents the results of the application of SAR images and different techniques for the assessment and the definition of landslides triggered by earthquake in the Mila regions (Algeria). The paper does not present relevant and particular novelties, relying on standard and very widely implemented applications such as Interferometric techniques (although with a newly developed algorithm such as LiCSBAS) and Coherence change detection; moreover, optical imagery were used to validate the results, however only through visual interpretation of pre-and post-event imagery. Moreover, the structure of the paper is not very clear and needs to be intensely revised. | As stated in the previous discussion, the authors will thoroughly revise the paper if the Editors request it. Evidently, the revised paper will incorporate all of the comments and suggestions made by the referees and readers. |
| II | Also, the authors should indicate which is the novelty of their work and how these standard approaches used are improved (if so). To strengthen the results obtained, the authors should consider also to use other SAR-based techniques, as amplitude analysis or pixel-offset techniques. | In this case, the use of the pixel-offset technique is limited due to the incoherent change of the ground. |
| III | Hereon, a list of detailed revisions to be addressed, in my opinion: The abstract needs to be revised in some points: what does exactly means disorder (in line 18 and 19)? Is there any geomorphological evidence? Please, use correct terminology to define these elements. | By disorder, we mean any ground changes. And, yes, there was geomorphological evidence in the area (see Figures 3 and 20). |
| IV | In line 21, please mention the exact number of interferograms used for the research. | As shown in table 2, 224 interferograms were used to perform time series analysis with LiCSBAS for this study, and we will add it in the phrase. |
| V | Line 23: is it real subsidence displacement or it is a deformation induced by landslide activity? please, specify and clarify it. | The subsidence deformation has occurred as a consequence of the movement of the main Kherba landslide. |
| VI | The introduction section is insufficient and does not provide a real comparison with the current state-of-the-art and does not highlight the achievements of this work and its novelty and added value in the current literature framework. Line 31: only the work cited highlight the usefulness of satellite imagery for prediction of landslides. Please mention additional works dealing with this topic and which different approaches can be mentioned. Lines 35-39: the sentence is very long and not completely clear. Please, consider to rewrite it. Moreover, provide additional and more updated literature. Line 41: please, specify what LiCSAR and LiCSBAS are. Section 2: the description of the study area is very weak and insufficient. Please, indicate the geological and geomorphological setting of the study area to fully characterize the deformational events occurring. Section 2.1 provides a sort of scheme of the research conducted. This could be summarized in the introduction or schematized in section 3. Line 70: is there any literature citing and describing the seismicity of the study area? Line 72: is there any existing landslide inventory of the study area? Section 2.3: please, consider to add a short description of the Sentinel-2 dataset. Line 94: why using this reference? It is not linked to the statement. Section 3.1: the description of the basic principles of SAR interferometry can be skipped, I would rather describe in a more specific way the LiCSAR and LiCSBAS software and approach. Moreover, if possible, please provide a workflow to summarize the approaches used in this research. Section 3.2: at the current state, this section is poorly described, without any specific indication on the technique used or on the dataset implemented. Section 4: Lines 168-171: please, consider to delete this paragraph, since it is a repetition of something already stated previously. Lines 172-178: please, use a conceptual scheme or a workflow to summarize what it is written here. Figure 5: please, consider to indicate LOS direction in the figures. Figure 7: please, consider to indicate LOS direction in the figures. | All comments and remarks raised by the referees and readers will be taken into account. |

| | | |
|---|---|---|
| | Figure 8: please, indicate dates of the several figures. Moreover, a better description of what can be observed in the figures should be provided within the text. As it is, the description in the text is insufficient.

Line 222: I would say that DInSAR has abundantly proved to be a solid technique for the monitoring of slow movements, not that is expected. Please, consider to use more up-to-date references.

Figure 9 and 10: please, consider to indicate LOS direction in the figures. For figure 10, please, indicate the biased pixels in the map.

Figure 11: the dates are rather difficult to be seen.

Figure 12: Please, indicate the dates and provide a better description of what can be seen here within the text.

In the CCD analysis, could you please also indicate the mean coherence value of the post-event phase? In this case, quantify the change as you have already done.

Section 4.2.2. This section is poorly described and in general the validation with S-2 images is insufficient. First of all, it is not comprehensible which kind of data treatment has been done. Thus, I do not see any particular change in the two images, as well as I do not see the cracks indicated and the motion direction. Please, consider to re-write and do again more specific analyses with optical imagery (e.g., change detection, specific codes, etc.).

Section 4.3: please, move the LiCSBAS description in section 3 (by adding some more details).

Discussion section: this section is very poor and does not provide any critical analysis of the results nor it is showing which is the novelty of this applications. Moreover, the latest point, related to the "new hillside deformation" should be clarified, improving the interpretation of this area.

Conclusions: Line 345: this statement is pretty obvious, InSAR is a consolidated technique which has continuously proved its efficacy over the last 30 years. | |
| VII | Line 35: InSAR is not an active sensor system, but a technique for the processing of SAR images. Please, use the proper terminology. | Indeed, the InSAR technique exploitation the active radar systems. |
| VIII | Line 49: how the analysis of the results can be used for early warning? Please, explain this statement. | Because of the results of the Grarem site analysis, which show some fringes and coherence loss that prove and indicate the slope's instability (no landslide occurred), which may be a sign of potential future land failure. This information obtained from the InSAR study can serve as early warning information. |
| IX | Line 59: what is a geotechnical disorder? Please, consider to use a more appropriate terminology. | We will replace it with a "geotechnical hazard"! |
| X | Figure 14 and 15: what the full lines are indicating? | Figure 14 depicts the frequency distributions of coherence values within the RoI, with the lines indicating the change in coherence over time. In Case A, the green line represents the pre-event coherence distribution, and the red line represents the post-event coherence distribution, which clearly shows a decay of the mean coherence after the main event (dates and values are presented in the legend).
Figure 15 illustrates why we chose the interferogram of the 22-28July (green line) as the pre-event (initial) even though there is another IFG of (22July-3Aug green dotted line) with only 4 days before the main event (7 August 2020). |
| XI | Please, highlight and describe better what can be seen outside of the landslide border, in particular along the SW flank, where considerable displacements are visible (Figure 17). If this is the area with possible "subsidence or landslides", how this can be interpreted? Can you rely on additional data to interpret the displacements? is it on a slope or on a flat area? can you estimate if the movement is vertical or horizontal? In | The region is located on the opposite hillside of the Kherba Hill, and both sides have a significant slope, so one possible explanation for this subsidence is probably related to the large mass movement of the |

| | | |
|---|---|---|
| | this case can you combine ascending and descending imagery to obtain vertical and horizontal projection from LOS data. | main landslide, causing the opposite side of the hill to move down (subsidence). |
| **XII** | Figure 18 and 19: first of all, these figures are poorly described within the text. Figure 18 is showing very noisy time series. Can you explain this? | It appears noisy due to the small number of IFGs used (short period 4 months before and two months after the main shock) compared to the analysis shown in figure 19. |

---

## Author Response (AR2)

**Report #1**

| | |
|---|---|
| Suggestions for revision or reasons for rejection (will be published if the paper is accepted for final publication)
I would like to thank the authors for improving the quality of the manuscript, following the indications provided by the reviewers. Despite this, I still have concerns, about the lack of innovation of the research, relying on standard applications and methodologies. In particular, the use of interferometric and coherence change techniques is a solid and standardized practice, and the assessment of the deformation through optical imagery is insufficient, without providing any quantitative information to be compared with the other sources.

Despite this, the case study is interesting. Another concern is related to the manuscript, which can be significantly improved: The introduction section is very synthetic and does not describe the importance of the research in the framework of the existing literature, as well as the discussion section is a mere geomorphological description of the deformational events and a summary of the results obtained. In this section, in particular, a critical analysis of the results should be given, comparing the results obtained with those of similar works, raising the significance and the innovation of the research, if any | All comments and remarks raised by the referee are taken into account. |

**Report #2**

| | |
|---|---|
| The manuscript presents an overview on the use of SAR-based interferometry, using ESA Sentinel 1 images, for landslide delineation and analysis. The Authors focus on some case studies located in north Algeria. Overall, the manuscript is interesting and clear, however, I think the main issues of this manuscript is its target, since it is not clear if it is the application of the methodology. In particular, the procedure used by Authors for image processing is described in details, whereas the analyses case studies seem to be simple examples. For this reason, the geological interpretation of the results is generic and poorly detailed. I would encourage the Authors to shorten the description of the methodology, enhancing the geological interpretation of results.
Finally, there is a very minor issue at page 1, row 27, where the Authors cite "Del and Idrogeologico, 2012", as well as in the reference list. I suppose the name of the Authors are wrong, as far as I deduced, this should be: "Mazzanti et al., 2012". Given the aforementioned points, I recommend to consider the manuscript after a moderate review. | All comments and remarks raised by the referees are taken into account and changed in the manuscript. |

---

## Author Response (AR3)

**Report #1**

| | |
|---|---|
| are there any other paper dealing with seismic-induced landslides monitoring? | Yes, some are mentioned in this paper (Goudarzi, 2010, Cascini et al., 2013; Wempen, 2020 , Jacquemart and Tiampo, 2021, Jacquemart and Tiampo, 2021,. Etc) but in this paper we investigate it using InSAR. |
| InSAR is good for that? | InSAR has abundantly proved to be a solid technique for monitoring such cases and slow movements. |
| Coherence analyses and optical imagery are a valid support? Is there literature supporting that? | Yes, coherence is a good method for detecting ground changes, but optical methods have some limitations. The Coherence method is used by Jacquemart and Tiampo, 2021, to detect landslides. |
| What's the added value given by your paper? | The work demonstrates the capability of using InSAR methods to monitor any land deformation, including landslides. |
| Was Licsbas technique already used? If so, in which other case studies? Is there any technical improvement by using Licsbas? | This technique has been used in time series estimation, and we cited some papers that used LiCSBAS in the references. |
| Was the use of open source software a valid support? | The main software used in this study was SNAP, which was developed by the European Space Agency (ESA). This software is widely used by scientists and has been mentioned in numerous papers. |
| There are many points raised which should be introduced and compared in the introduction and discussion sections, otherwise I cannot understand why you are writing this paper and which is the main aim of it. | We investigated the area by InSAR in the aftershock of an earthquake in order to identify geotechnical displacements or any land deformations, their extent, and locations. We have two cases in this area, one has occurred, and the other (Grarem) is suspected to be an imminent landslide due to the same conditions. |
| Finally, I think the title should be revised highlighting that this is a case study, not a general paper on seismic-induced landslide monitoring by InSAR (which is also not the only technique used here). | InSAR is the primary methods from which other are derived technique. |

---

## Author Response (AR4)

**Report #4**

| | |
|---|---|
| Please ensure that the colour schemes used in your maps and charts allow readers with colour vision deficiencies to correctly interpret your findings. | We had changed some charts styles and their lines to be clear and readers can distinguish between. |
| My last suggestion is to insert the name of the case study in the title (as also suggested by one of the referee). | For the title: **"Earthquake-induced landslides monitoring and survey by means of InSAR. Case of the 2020 Mila earthquake"** |